# Combined Effect of Naturally-Derived Biofilm Inhibitors and Differentiated HL-60 Cells in the Prevention of *Staphylococcus aureus* Biofilm Formation

**DOI:** 10.3390/microorganisms8111757

**Published:** 2020-11-09

**Authors:** Inés Reigada, Clara Guarch-Pérez, Jayendra Z. Patel, Martijn Riool, Kirsi Savijoki, Jari Yli-Kauhaluoma, Sebastian A. J. Zaat, Adyary Fallarero

**Affiliations:** 1Drug Research Program, Division of Pharmaceutical Biosciences, Faculty of Pharmacy, University of Helsinki, FI-00014 Helsinki, Finland; kirsi.savijoki@helsinki.fi (K.S.); adyary.fallarero@helsinki.fi (A.F.); 2Department of Medical Microbiology and Infection Prevention, Amsterdam institute for Infection and Immunity, Amsterdam UMC, University of Amsterdam, 1105 AZ Amsterdam, The Netherlands; c.m.guarchperez@amsterdamumc.nl (C.G.-P.); m.riool@amsterdamumc.nl (M.R.); s.a.zaat@amsterdamumc.nl (S.A.J.Z.); 3Drug Research Program, Division of Pharmaceutical Chemistry and Technology, Faculty of Pharmacy, University of Helsinki, FI-00014 Helsinki, Finland; jayendra.patel@helsinki.fi (J.Z.P.); jari.yli-kauhaluoma@helsinki.fi (J.Y.-K.)

**Keywords:** *Staphylococcus aureus*, biomaterials, medical devices, HL-60 cells, PMNs, biofilm, endotracheal tube, titanium, implantable devices, nosocomial diseases

## Abstract

Nosocomial diseases represent a huge health and economic burden. A significant portion is associated with the use of medical devices, with 80% of these infections being caused by a bacterial biofilm. The insertion of a foreign material usually elicits inflammation, which can result in hampered antimicrobial capacity of the host immunity due to the effort of immune cells being directed to degrade the material. The ineffective clearance by immune cells is a perfect opportunity for bacteria to attach and form a biofilm. In this study, we analyzed the antibiofilm capacity of three naturally derived biofilm inhibitors when combined with immune cells in order to assess their applicability in implantable titanium devices and low-density polyethylene (LDPE) endotracheal tubes. To this end, we used a system based on the coculture of HL-60 cells differentiated into polymorphonuclear leukocytes (PMNs) and *Staphylococcus aureus* (laboratory and clinical strains) on titanium, as well as LDPE surfaces. Out of the three inhibitors, the one coded **DHA1** showed the highest potential to be incorporated into implantable devices, as it displayed a combined activity with the immune cells, preventing bacterial attachment on the titanium and LDPE. The other two inhibitors seemed to also be good candidates for incorporation into LDPE endotracheal tubes.

## 1. Introduction

Over 2.6 million new cases of healthcare-associated infections are annually reported just in the European Union [1], and over 33,000 result in death [2] due to the increasing number of antimicrobial-resistance cases [3]. At least 25% of these infections are associated with the use of medical devices, and 80% of them are estimated to be caused by bacterial biofilms [4,5]. Biofilms are defined as a community of microorganisms encased within a self-produced matrix that adheres to biological or nonbiological surfaces [6,7]. They are currently regarded as the most important nonspecific mechanism of antimicrobial resistance [8,9].

During the worldwide crisis of SARS-CoV-2, as well as in many other pathological conditions, mechanical ventilation is used to assist or replace spontaneous breathing as a life-saving procedure in intensive care. However, the use of endotracheal intubation also poses major risks in prolonged ventilation. The endotracheal tube provides an ideal opportunity for bacteria to form biofilms on both the outer and luminal surface of the tube, increasing the risk of pulmonary infection by 6 to 10 times [10,11,12], with *Staphylococcus* spp. and *Pseudomonas aeruginosa* among the most frequent colonizing agents [13]. Colonization by microorganisms and the subsequent formation of a biofilm can happen within hours [13], but these kinds of devices are relatively easy to replace. On the other hand, infection of orthopedic implants is particularly problematic, as these devices remain in the body, often causing chronic and/or recurring infections mediated by biofilms. These infections also frequently require removal of the infected implant, thereby causing implant failure [14,15,16]. Given the rising number of implantations, the absolute number of complications is inevitably increasing at the same pace, causing not only distress for the patients but also an increasing economic burden [15,16].

The most common causative agents of infection in orthopedic implants are Gram-positive cocci of the genus *Staphylococcus,* e.g., *Staphylococcus aureus* and *Staphylococcus epidermidis* [17]. In the absence of a foreign body, contaminations caused by these opportunistic pathogens are usually cleared by the immune system. In contrast, the placement of an implant per se represents a risk factor for the development of a chronic infection. This is due to the fact that the surgical procedure causes tissue damage resulting in the local generation of damage-associated molecular patterns (DAMPs), endogenous danger molecules that are released from damaged or dying cells and activate the innate immune system by interacting with pattern recognition receptors (PRRs) [18]. This is sensed by host neutrophils, which migrate to the injured tissue sites, activating defense mechanisms, such as the generation of oxygen-derived and nitrogen-derived reactive species as well as phagocytosis, to unsuccessfully attempt to clear the foreign material. These events lead to immune cell exhaustion and death, and tissue damage caused by the triggered inflammation eventually leads to a niche of immune suppression around the implant [19]. Under these specific conditions, the clearance of planktonic bacteria by immune cells becomes impaired [14], which predisposes the implant to microbial colonization and biofilm-mediated infection.

However, it is possible that not only host immune cells and bacterial cells can be present at the moment of implantation, but also the cells of the tissue where the material is being implanted. The “race for the surface” concept describes the competition between bacteria and the host cells of the tissue where the device is implanted to colonize the surface of the material [20]. The rapid integration of the material into the host tissue is a key component in the success of an implant, as the colonization of the surface by the cells of the host not only ensures correct integration, but also prevents bacterial colonization [21]. However, if bacterial adhesion occurs first, the host defense system is unable to prevent the colonization and subsequent biofilm formation [22].

One of the current challenges to prevent and resolve biofilm-mediated infections is the limited repertoire of compounds that are able to act on them at sufficiently low concentrations [23]. Because of this, there is intense ongoing research focused on the search for new biofilm inhibitors by means of chemical screenings. However, for compounds to be truly effective when used for the protection of biomaterials in translational applications, they must be tested in meaningful experimental models. Based on this, we previously optimized an in vitro system based on the coculture of SaOS-2, osteogenic cells, and *S. aureus* (laboratory and clinical strains) on a titanium surface (Reigada; et al. [24]), and studied the effect of three naturally derived biofilm inhibitors (Figure 1). Two of them were dehydroabietic acid (DHA) derivatives, namely, *N*-(abiet-8,11,13-trien-18-oyl)cyclohexyl-L-alanine and *N*-(abiet-8,11,13-trien-18-oyl)-D-tryptophan, coded **DHA1** (Figure 1a) and **DHA2** (Figure 1b), respectively. The third one was a flavan derivative, 6-chloro-4-(6-chloro-7-hydroxy-2,4,4-trimethylchroman-2-yl)benzene-1,3-diol, coded **FLA1** (Figure 1c). All of them were previously reported by our group and demonstrated to display activity in preventing biofilm formation, as well as disrupting preformed *S. aureus* biofilms on 96-well plates [25,26], but the testing in the coculture model with osteogenic cells provided with new insights into their applicability as part of anti-infective implantable devices.

The coculture model developed previously [24] involving SaOS-2 cells and *S. aureus* strains offered an in vitro environment that was closer in terms of host–bacteria interactions and substrate materials to the in vivo scenario of the implanted device. Moreover, in terms of antimicrobial evaluation, it provided information not only regarding the antibiofilm activity but also about the effects on tissue integration. However, this model did not assess the effect of the antimicrobials on immune cells. As mentioned earlier, chronic inflammation not only lowers the antimicrobial efficacy, but also complicates the integration of the implant material as a consequence of maintained inflammation and tissue damage. Therefore, it is essential to assess the effect that antimicrobials might have in the presence of immune cells, particularly for those intended to be used in medical devices. Endotracheal tubes significantly differ from orthopedic implants in material, function, and implantation procedure, but they both cause impairment of the host immune antimicrobial capacity.

Because of this, in this contribution, we move one step forward toward emulating in vivo conditions encountered by medical devices in an in vitro setting. In this case, we introduce immune cells, specifically neutrophils, in a coculture model with bacterial cells. Neutrophils were selected as immune cells as they are the first cell types to migrate toward damaged tissue cells [27]. Our aim was to develop cocultures of *S. aureus* and differentiated HL-60 cells, grown on titanium coupons or low-density polyethylene (LDPE) tubes, to simulate biofilm formation on orthopedic implants or endotracheal tubes, respectively. As a proof-of-concept, we also study the possible antimicrobial effects of naturally derived antibiofilm inhibitors **DHA1**, **DHA2**, and **FLA1**, and their possible applicability as part of medical devices.

## 2. Materials and Methods

### 2.1. Compounds

The dehydroabietic acid derivatives coded **DHA1** and **DHA2** (previously coded 11 and 9b, respectively) were synthesized according to [25]. Their spectral data were identical to those reported in [28]. The flavan derivative coded **FLA1** (previously coded 291 in [26]), was purchased from TimTec (product code: ST075672, www.timtec.net). These compounds were selected given their low minimum inhibitory concentrations (MICs) and the low concentrations needed in order to prevent *S. aureus* biofilm formation (Appendix A). The control antibiotic rifampicin was purchased from Sigma-Aldrich (CAS number 13292-46-1) and coded **RIF**. Molecular structures are shown in Figure 1.

### 2.2. Bacterial Strains

Bacterial studies were performed using the laboratory strain *S. aureus* ATCC 25923 (American Type Culture Collection, Manassas, VA, USA) and one clinical strain isolated from hip prostheses and osteosynthesis implants at the Hospital Fundación Jiménez Díaz (Madrid, Spain) [29] (coded *S. aureus* P2).

### 2.3. HL-60 Cell Culture and Differentiation

HL-60 (ATCC CCL-240) cells were grown and maintained in Roswell Park Memorial Institute (RPMI) 1640 Medium (R8758, Sigma Aldrich, St. Louis, MO, USA), supplemented with 20% (*v*/*v*) heat-inactivated fetal bovine serum (FBS) (Sigma Aldrich, St. Louis, MO, USA) and 1% (*v*/*v*) penicillin/streptomycin (Sigma Aldrich, St. Louis, MO, USA). Cells were maintained in suspension at a concentration between 10^5^–10^6^ cells/mL in 72 cm^2^ culture flasks (VWR, Radnor, PA, USA) at 37 °C in 5% CO_2_ in a humidified incubator. *N,N*–Dimethylformamide (DMF)(Sigma Aldrich, St. Louis, MO, USA) was utilized in order to differentiate the cells into polymorphonuclear-like cells [30]. In order to carry out the differentiation, cells were cultured for 6 days in the maintenance medium at a concentration of 100 mM DMF. The success of the differentiation was assessed visually after Giemsa staining using a Leica DMLS microscope (Leica, Wetzlar, Germany). Differentiated cells were neutrophil-like, with a multilobar nucleus and a fairly clear cytoplasm.

### 2.4. Biofilm Prevention Efficacy of Differentiated HL-60 Cells against Different Bacterial Concentrations of S. aureus ATCC 25923

#### 2.4.1. Bacterial Inoculum Preparation

Pure colonies (2–3) of *S. aureus* ATCC 25923 were added to 5 mL of tryptic soy broth (TSB, Neogen^®^, Lansing, MI, USA) and incubated overnight at 37 °C, 120 rpm. After incubation, 10 µL of the preculture was added to 5 mL fresh TSB and incubated for 3–4 h (37 °C, 120 rpm), in order to obtain a midlogarithmic growth phase of bacteria. Bacterial cultures were washed twice with sterile phosphate-buffered saline (PBS; 140 mM NaCl, pH 7.4) and the concentration was adjusted to 2 × 10^8^ in RPMI 1640 based on the optical density of the suspension at 600 nm. From this stock, serial dilutions were made in RPMI 1640 between 2 × 10^8^ and 2 × 10^2^ CFU/mL.

#### 2.4.2. Immune Cell (HL-60) Preparation

Twenty-four hours prior to starting the experiments, the media of the differentiated HL-60 cells were refreshed with nonsupplemented RPMI 1640 in order to remove possible traces of antibiotics from the maintenance media. Differentiated HL-60 cells (after 6 days exposure to a concentration of 100 mM DMF) were counted with a Countess™ II automated cell counter (Thermo Scientific, Waltham, MA, USA) and adjusted to a concentration of 2 × 10^5^ cells/mL. In order to test the influence of activating the cells with phorbol 12-myristate 13-acetate (PMA) (Sigma Aldrich, St. Louis, MO, USA) on the antimicrobial capacity, half of the suspension of differentiated HL-60 cells was incubated for 20 min in RPMI 1640 supplemented with 25 nM PMA. PMA-activated and nonactivated HL-60 cells were separately added (100 µL) to flat-bottomed, 96-well microplates (Nunclon Δ surface, Nunc, Roskilde, Denmark).

#### 2.4.3. Coculture of *S. aureus* ATCC 25923 and HL-60 Cells

*S. aureus* ATCC 25923 suspensions at the different concentrations (100 µL) were added to the wells of the 96-well microplate already containing 100 µL of one of the two different HL-60 cell suspensions (activated or nonactivated). In the bacterial control wells, 100 µL of the *S. aureus* ATCC 25923 suspension at different concentrations were added to wells containing 100 µL of RPMI 1640 alone. The wells corresponding to the HL-60 cell control consisting of a suspension of HL-60 cells at a concentration 10^5^ cells/mL in RPMI 1640 were used to observe the cell morphology after 18 h of incubation, but no quantitative viability test was carried out. The 96-well microplates were incubated for 18 h at 37 °C and 5% CO_2_ in a humidified incubator.

#### 2.4.4. Biofilm Quantification in 96-Well Microplates

After *S. aureus* ATCC 25923 biofilms were grown, the media were carefully removed and each well was washed twice with 125 µL sterile PBS, followed by the addition of 150 µL of TSB. Each 96-well microplate was closed with a plastic seal and parafilm and placed in a plastic bag, which was sealed with heat in order to prevent leakage. The plate was sonicated for 10 min at 35 kHz in an Ultrasonic Cleaner 3800 water bath sonicator (Branson Ultrasonics, Danbury, CT, USA) at 25 °C. This procedure did not affect the viability of the staphylococci (Appendix A).

*S. aureus* ATCC 25923 was then resuspended in RPMI 1640 using 3 pipetting cycles (up/down). Samples (10 µL) from each tested condition were transfer to 90 µL of TSB and serial dilutions were made from 10^−1^ up to 10^−7^. Aliquots (10 µL) of all dilutions were transferred to sheep blood agar plates (Amsterdam UMC, Amsterdam, The Netherlands) and incubated at 37 °C overnight. Viable plate colonies were counted the next day.

### 2.5. Influence of Opsonizing S. aureus ATCC 25923 on the Efficacy of HL-60 Cells in Preventing Bacterial Attachment on Titanium Coupons

*S. aureus* ATCC 25923 suspensions were prepared as described in Section 2.4.1. The inoculum was adjusted to 2 × 10^7^ CFU/mL and opsonized using 50% (*v*/*v*) pooled human serum in PBS (pooled from 15 healthy blood donors) by incubating at 37 °C for 30 min with gentle agitation, washing twice with PBS, and resuspending in RPMI 1640 [31]. As a control, a nonopsonized suspension of *S. aureus* ATCC 25923 was used. The two different suspensions (opsonized and nonopsonized), were then added to sterilized titanium coupons (0.4 cm height, 1.27 cm diameter, BioSurface Technologies Corp., Bozeman, MT, USA).

On the other hand, HL-60 cells were prepared as described in Section 2.4.2 and added to the titanium coupons, to which the bacterial suspension was previously added. The final volume covering each coupon was 1 mL, made up of 500 µL of a suspension of 2 × 10^7^ CFU/mL *S. aureus* ATCC 25923, and 500 µL of a suspension of 2 × 10^5^ HL-60 cells/mL in RPMI 1640.

### 2.6. Effect of the Antimicrobial Compounds on the Prevention of S. aureus ATCC 25923 and S. aureus P2 Adhesion in Coculture with Differentiated HL-60 Cells on Titanium Coupons

#### 2.6.1. Culture of Staphylococci and HL-60 Cells

Bacterial inocula of *S. aureus* ATCC 25923 and *S. aureus* P2 were prepared as described in Section 2.4.1. The concentration was adjusted to 2 × 10^7^ CFU/mL in RPMI 1640. For the differentiated HL-60 cells, the media were refreshed with RPMI 1640 24 h before the experiments started in order to remove possible traces of antibiotics from the maintenance media. On the day of the experiment, cells were counted with a Countess™ II automated cell counter (Thermo Scientific, Waltham, MA, USA) and adjusted to a concentration of 2 × 10^5^ cells/mL in RPMI 1640. Each suspension (bacterial and HL-60 cells suspensions, 500 µL of each) was added to sterilized titanium coupons that were inserted in the different wells of a 24-well plate (Nunclon Δ surface, Nunc, Roskilde, Denmark) and to which the different tested compounds or the control antibiotics were added at a concentration of 50 µM (0.25% DMSO).

The 24-well plates containing the titanium coupons were maintained in cocultures with a solution containing 10^7^ CFU of *S. aureus* ATCC 25923 or *S. aureus* P2 and 10^5^ HL-60 cells in a total volume of 1 mL of RPMI 1640 for 24 h. Titanium coupons with added **RIF** (50 μM, 0.25% DMSO) were used as positive antibiotic controls. As coculture controls, titanium coupons without the addition of the tested antimicrobial compounds or control antibiotics were exposed simultaneously to both cellular systems (*S. aureus* and HL-60 cells) at the concentrations previously described. In addition, bacterial controls (exposed or not to the antimicrobial compounds in the absence of differentiated HL-60 cells) were also included.

#### 2.6.2. Bacterial Adherence on Titanium Coupons

Titanium coupons were gently washed with TSB to remove remaining adhering planktonic cells and transferred to Falcon tubes containing 1 mL of 0.5% (*w*/*v*) Tween^®^ 20-TSB solution. Next, the tubes were sonicated in an Ultrasonic Cleaner 3800 water bath sonicator (Branson Ultrasonics, Danbury, CT, USA) at 25 °C for 5 min at 35 kHz. The tubes were vortexed for 20 s prior to and after the sonication step. Serial dilutions of the resulting bacterial suspensions were made from 10^-1^ up to 10^−7^ and plated on tryptic soy agar (Neogen^®^, Lansing, MI, USA) plates.

### 2.7. Effect of the Antimicrobial Compounds on the Prevention of S. aureus ATCC 25923 Adhesion in Coculture with Differentiated HL-60 Cells on LDPE Tubes

The assay was carried out as described in Section 2.6 but instead of titanium coupons the materials added to the 24-well plates were 1 cm long sections of a sterilized fine bore LDPE tubing (Smiths Medical ASD, Minneapolis, MN, USA).

### 2.8. Scanning Electron Microscopy (SEM)

In order to visualize the effect of the antimicrobial compounds on the prevention of *S. aureus* ATCC 25923 adhesion in coculture with immune cells on titanium coupons, *S. aureus* ATCC 25923 was cocultured with differentiated HL-60 cells on titanium coupons, as described in Section 2.6. Prior to SEM, samples were fixed in 4% (*v*/*v*) paraformaldehyde and 1% (*v*/*v*) glutaraldehyde (Merck, Kenilworth, NJ, USA) overnight at room temperature. Samples were rinsed twice with distilled water, with the duration of each cycle being 10 min. The dehydration procedure consisted of 2 cycles of incubation for 15 min in 50% (*v*/*v*) ethanol, 2 cycles of incubation for 20 min in 70% (*v*/*v*) ethanol, 2 cycles of incubation for 30 min in 80% (*v*/*v*) ethanol, 2 cycles of incubation for 30 min in 90% (*v*/*v*) ethanol, and 2 cycles of incubation for 30 min in 100% (*v*/*v*) ethanol. In order to reduce the sample surface tension, samples were immersed in hexamethyldisilazane (Polysciences Inc., Warrington, FL, USA) overnight and air-dried. Before imaging, samples were mounted on aluminum SEM stubs and sputter-coated with a 4 nm platinum–palladium layer using a Leica EM ACE600 sputter coater (Leica Microsystems, Wetzlar, Germany). Images were acquired at 2 kV using a Zeiss Sigma 300 SEM (Zeiss, Oberkochen, Germany) at the Electron Microscopy Center Amsterdam (Amsterdam UMC).

Of each coupon, 8–10 fields were viewed and photographed at magnifications of 250×, 500×, 1000×, and 3000×. Representative images are shown in the results.

### 2.9. Statistical Analysis

The quantitative data are reported as the mean and standard deviation (SD) of at least three independent experiments. Data were analyzed using GraphPad Prism 8 for Windows. For statistical comparisons, Tukey’s multiple comparison test and Welch’s unpaired *t*-test were applied, and *p* < 0.05 was always considered as statistically significant.

## 3. Results

### 3.1. Effect of PMA Activation of Differentiated HL-60 Cells on Prevention of S. aureus ATCC 25923 Biofilm Formation

Before performing the experiments with the titanium coupons and the antimicrobial compounds, the initial concentration of *S. aureus* ATCC 25923 was determined where the bacteria were able to form a biofilm in absence of the HL-60 cells, but were prevented from forming a biofilm when cocultured with 10^5^ HL-60 cells. At the same time, it was assessed whether the activation of the HL-60 cells with PMA enhanced their bacterial clearance capability (Figure 2). At an initial *S. aureus* ATCC 25923 concentration of 10^8^ CFU/mL, HL-60 cells did not significantly affect the bacterial attachment in 96-well microplates. A reduction on the adhered *S. aureus* ATCC 25923 viable cell counts was observed at an *S. aureus* ATCC 25923 concentration of 10^7^ CFU/mL and below (*p* < 0.001 in all cases when comparing the bacterial control with bacteria cocultured with HL-60 cells, and *p* = 0.001 for cells activated with PMA and a starting inoculum of 10^3^ CFU/Ml), with the exception of the lowest bacterial concentration tested, i.e., 10^2^ CFU/mL, where no difference was found. Both PMA-activated and nonactivated HL-60 cells (gray and green columns, respectively) showed similarly reduced numbers of adherent *S. aureus* ATCC 25923 viable cell counts at 24 h, so it was concluded that activation with PMA does not significantly enhance *S. aureus* ATCC 25923 clearance.

Based upon these results, the optimal initial *S. aureus* ATCC 25923 concentration was found to be 10^7^ CFU/mL, which was then selected for the rest of the experiments. Since PMA activation of the HL-60 cells did not influence their phagocytic activity, no PMA stimulation was performed in subsequent experiments.

### 3.2. Influence of Opsonizing S. aureus ATCC 25923 on the Efficacy of HL-60 Cells in Preventing Bacterial Attachment on Titanium Coupons

One of the most relevant mechanisms of the host defense against *Staphylococcus* spp. is phagocytosis. Given that opsonization of *S. aureus* is important for neutrophils to be able to clear planktonic *S. aureus* [32], the phagocytic efficacy of HL-60 cells, as well as the impact of opsonization of *S. aureus* ATCC 25923, in preventing *S. aureus* ATCC 25923 biofilm formation on titanium coupons was simultaneously explored (Figure 3). The left part of Figure 3 shows how the preventive capability of HL-60 cells was preserved when tested on titanium surfaces (*p* < 0.001, when comparing *S. aureus* + HL-60 (gray column) with the bacterial control (black column)). This bacterial clearance activity was also observed when *S. aureus* ATCC 25923 was opsonized (*p* = 0.007), but no significant differences were found when comparing the effects of the HL-60 cells on opsonized versus nonopsonized *S. aureus* ATCC 25923 (*p* = 0.260). The antibacterial effects of PMA-activated HL-60 cells against opsonized *S. aureus* ATCC 25923 was additionally explored, but no differences were found with the antimicrobial activity of nonactivated HL-60 against nonopsonized *S. aureus* ATCC 25923 (Appendix A). Because opsonization did not enhance the bacterial clearance capacity of HL-60, the use of opsonized *S. aureus* during the rest of the experiments was decided against.

### 3.3. Effect of the Antimicrobial Compounds on the Prevention of S. aureus Adhesion in Coculture with Differentiated HL-60 Cells on Titanium Coupons

The effects of the three biofilm inhibitors **DHA1**, **DHA2**, and **FLA1** and one control antibiotic, **RIF**, on the prevention of *S. aureus* attachment on titanium coupons were investigated using two strains, the laboratory strain ATCC 25923 (Figure 4a) and the clinical isolate P2 (Figure 4b), either in the absence (gray bars) or presence of HL-60 cells (green bars). In the absence of HL-60 cells, all of the tested antimicrobial compounds, as well as the control antibiotic, significantly reduced the attachment of *S. aureus* ATCC 25923 (*p* < 0.001, *p* = 0.040, *p* < 0.001, and *p* < 0.001, for **DHA1**, **DHA2**, **FLA1**, and **RIF** versus the control, respectively). In the case of the clinical strain (Figure 4b), all the antimicrobial compounds, except **DHA2**, also showed antimicrobial activity in the absence of HL-60 cells (*p* = 0.0067, *p* < 0.001, and *p* = 0.0087 versus control for **DHA1**, **FLA1**, and **RIF**, respectively).

The right part of Figure 4a (green bars) corresponds to the same experiments performed in the presence of HL-60 cells. Under such conditions, there was also a significant reduction of the attached viable *S. aureus* ATCC 25923 when compared with the material incubated without HL-60 cells (green control bar versus black control bar, *p <* 0.001). Moreover, this reduction was further increased by compound **DHA1** (*p* = 0.025 when comparing the coculture control column with the **DHA1** column in the HL-60 cells + *S. aureus* ATCC 25923 section of Figure 4, and *p* < 0.001 when comparing the **DHA1** gray column with the **DHA1** green column). The same tendency was observed for **DHA2**, but no statistical differences were found when comparing this to the control (dark green control: HL-60 cells + *S. aureus* ATCC 25923 section of Figure 4). Despite the fact that **FLA1** and the model antibiotic (**RIF**) successfully prevented the adhesion of *S. aureus* ATCC 25923, they did not cause a further increase in the bacterial clearance activity of the HL-60 cells. In contrast, the mere presence of HL-60 cells did not result in a significant reduction of *S. aureus* P2 attachment (Figure 4b), since no differences were found between the *S. aureus* P2 control in monoculture and in coculture with HL-60 cells. In this case, no differences were found between the antibacterial effects of the compounds in monoculture when compared with the same treatment in coculture with HL-60 cells (**DHA1**, **DHA2**, **FLA1**, and **RIF** gray columns versus their corresponding green columns).

Using SEM imaging, it was visually confirmed that **DHA1** reduced the number of *S. aureus* ATCC 25923 adherent to the titanium surface (Figure 5). In fact, almost no cocci were observed on the **DHA1-**treated titanium across the entire coupon. In addition, the presence of HL-60 cells also reduced the bacterial attachment, which was further enhanced by the treatment of **DHA1**. Adherent HL-60 cells were observed, as seen in the last row of images.

### 3.4. Effect of the Antimicrobial Compounds on the Prevention of S. aureus ATCC 25923 Adhesion in Coculture with Differentiated HL-60 Cells on LDPE tubes

The adhesive capacity of *S. aureus,* as well as the antimicrobial effect of different compounds, is known to significantly differ depending on the material [33]. For this reason, the applicability of the compounds as part of endotracheal tubes was tested on a clinically relevant material, LDPE. Figure 6 shows the effects of the tested compounds and the control antibiotic (**RIF**) on the prevention of *S. aureus* ATCC 25923 attachment on LDPE tubes. The left part of the figure shows that all the antimicrobial compounds significantly reduced the numbers of attached viable *S. aureus* ATCC 25923 cells, in the absence of HL-60 (*p <* 0.001, *p* = 0.0035, *p* < 0.001, and *p* < 0.001, when comparing, respectively, **DHA1**, **DHA2**, **FLA1**, and **RIF** with the control). Adding HL-60 cells resulted in significant prevention of *S. aureus* ATCC 25923 attachment to LDPE tubes (*p* < 0.001 when comparing the coculture control (dark green column) with the bacterial control (black column)). In this case, all the antimicrobial compounds looked to be able to further potentiate this bacterial clearance capability (*p* < 0.001, *p* = 0.023, *p* = 0.002, and *p* < 0.001, when comparing, respectively, **DHA1**, **DHA2**, **FLA1**, and **RIF** light green columns with the control dark green column). However, similarly to what was observed on the titanium model, it was only the **DHA1** treatment that further potentiated the action of HL-60 cells against *S. aureus* ATCC 25923 (*p* = 0.015, when comparing the **DHA1** gray column with the **DHA1** green column). No differences were found between the viable cells (CFU/mL) attached on the LDPE exposed to *S. aureus* ATCC 25923 and treated with **DHA2**, **FLA1**, and **RIF** in monoculture and those exposed to both *S. aureus* ATCC 25923 and HL-60 with the same treatments (gray columns versus green columns). Full bacterial clearance was detected in LDPE tubes in the presence of **DHA1** and **RIF** (i.e., no viable bacterial counts measured), where cocultures of *S. aureus* ATCC 25923 with differentiated HL-60 cells were formed. These findings further highlight the relevance of **DHA1** as an antimicrobial candidate for incorporation into medical devices.

## 4. Discussion

In this study, we explored the potential of incorporation of three previously identified naturally derived biofilm inhibitors into medical devices, particularly for titanium implantable devices and LDPE endotracheal tubes. From the three tested antimicrobial compounds, in line with previous findings [24], **DHA1** appeared to be the best candidate for incorporation as part of implantable medical devices. All of the compounds proved to be interesting candidates to include into anti-infective endotracheal tubes, but it was **DHA1** that again showed itself to be the most promising candidate, as it was the only one that significantly increased the bacterial clearance capacity of HL-60 cells against *S. aureus* ATCC 25923.

For compounds to be truly effective when used for protection of biomaterials in translational applications, they must be tested in meaningful experimental models. The insertion of any device provokes an acute inflammatory response that may cause ineffectiveness of innate immune cells such as neutrophils in cleaning planktonic bacteria, since these cells are directed to degrade the material. For this reason, it is vitally important to study the effects of antimicrobial compounds on neutrophils to assess their suitability as part of medical devices.

Before this investigation, no published reports existed on the effect of the two DHA derivatives (**DHA1** and **DHA2**) on neutrophils, however, the parent compound (dehydroabietic acid, DHA) was previously reported to have slight toxicity toward this cell type [34]. This prior knowledge further justified the need for an assessment of the effects of **DHA1** and **DHA2** on the bacterial clearance capacity of neutrophils. Similarly, no data existed on effects of **FLA1** on neutrophils, but other flavan derivatives were studied. Out of 10 different flavan-3-ol derivatives tested on human neutrophils, only two presented a slight toxic effect toward the neutrophils, but all of them reduced reactive oxygen species (ROS) and interleukin-8 production [35]. On the other hand, flavan-3-ol derivatives extracted from *Bistorta officinalis* (Delarbre) were reported to inhibit tumor necrosis factor-α (TNF-α) release from neutrophils [36]. These earlier findings were promising in terms of applying the flavan derivative **FLA1** as part of medical devices, since its antimicrobial capacity in combination with its anti-inflammatory effects could result in prevention of infection while providing ideal cues toward material integration and resolution of inflammation.

In this study, we utilized HL-60 cells differentiated to polymorphonuclear-like cells in order to study the effects of the antimicrobial compounds in the presence of neutrophils. Alternatively, freshly extracted neutrophils could also have been used, but in such case differences would be encountered between individual donors in terms of reproducibility, the total number of cells that can be harvested, their short lifespan, or the disturbances in their physiology due to isolation procedures [37,38].

The three tested antimicrobial compounds and the control antibiotic (**RIF**) reduced *S. aureus* ATCC 25923 adhesion to titanium in the absence of HL-60 at the tested concentration of 50 µM. In the case of the clinical *S. aureus* strain, all compounds, except **DHA2**, significantly reduced *S. aureus* P2 biofilm formation on titanium in the absence of HL-60. The prevention of biofilm formation by **DHA1** and **FLA1** was as expected, as the compounds were used at concentrations higher than their MIC values [25,26]. Compound **DHA2** showed some prevention activity, despite being tested at a concentration slightly below its MIC (i.e., 60 µM) [26]. As with most antimicrobials, cytotoxicity is a concern, and **DHA2** was shown to reduce viability of HL cells (originating from the human respiratory tract) at a concentration of 100 µM [25].

The mere presence of HL-60 cells significantly reduced the bacterial attachment of *S. aureus* ATCC 25923. In contrast, this effect was not observed for the clinical *S. aureus* strain P2. This was also observed in our previous study, with the results obtained with the laboratory strain *S. aureus* ATCC 25923 significantly differing from the ones obtained with *S. aureus* P2 under the same experimental conditions. The latter has a key relevance in assessing the applicability of biofilm inhibitors as part of titanium implantable devices, as it was isolated from patients with orthopedic device-related infections. For this reason, the additional measurement of the preventive capacity of the biofilm inhibitors against the clinical *S. aureus* P2 strain is of great relevance.

Compound **DHA1** was shown to further potentiate *S. aureus* ATCC 25923 clearance caused by HL-60, and this effect was further confirmed by SEM. This compound also managed to effectively prevent the adherence of the clinical strain (*S. aureus* P2), but in this case it was difficult to establish if the reduction in bacterial attachment caused by **DHA1** (as well as **FLA1** and **RIF**) was due to a combined antimicrobial effect with the HL-60 cells or if it was due to their intrinsic antimicrobial capacity, as their effects in monoculture were equal to those observed in coculture with the HL-60 cells. These results emphasized the importance of not limiting the in vitro experimentation to laboratory strains, especially in cases aimed at finding compounds effective against medical device-associated infections, as the results obtained with laboratory strains may overestimate the efficacy of the compound.

These results demonstrate that **DHA1** seems to further increase *S. aureus* ATCC 25923 clearance by HL-60, while none of the other compounds negatively affect the antimicrobial effect of these immune cells. This is also of high relevance, as adverse effects on the immune response could be detrimental, and even increase the risk of infection. As an example, Croes et al. [39] biofunctionalized the surface of titanium implants with chitosan-based coatings that were incorporated with different concentrations of silver nanoparticles. Despite the good antimicrobial results obtained in the in vitro tests, these coatings did not demonstrate antibacterial effects *in vivo.* Due to the toxicity of the silver nanoparticles on the immune cells, these coatings aggravated infection-mediated bone remodeling, including increased osteoclast formation and inflammation-induced new bone formation.

Similar results were obtained in LDPE tubes, with the bacterial clearance capacity of the HL-60 cells against *S. aureus* ATCC 25923 also observed, and **DHA1** further potentiated this activity. The prevention of *S. aureus* adherence on the surface of endotracheal tubes may have potential to significantly reduce the rates of ventilator-associated pneumonia caused by these bacteria [40]. Additionally, by preventing the attachment of *S. aureus*, the attachment of *P. aeruginosa* may also be hampered, as several infection models demonstrated how early colonization by *S. aureus* facilitated subsequent *P. aeruginosa* colonization [41,42]. The development of a dual-species biofilm is expected to not only strongly worsen the pathology but significantly complicate the treatment [43].

In this study, and in concordance with our previous findings, **DHA1** was identified as the best candidate to be incorporated into implantable devices. This is because, in addition to its intrinsic antibiofilm capacity against both laboratory and clinical strains of *S. aureus*, it also seems to be able to enhance *S. aureus* ATCC 25923 clearance by HL-60 cells. Further mechanistic studies should be performed in order to elucidate if **DHA1** has a direct effect on the antimicrobial activity of PMNs. In the near future, we plan to assess the effects of **DHA1** on phagocytosis, ROS production, and formation of neutrophil extracellular traps (NETs).

Given the promising results that **DHA1** showed, both in our previous publications and in the current one, this biofilm inhibitor is involved in plans to be integrated as part of a titanium coating by means of 3D printing, with the coating formulation consisting of **DHA1**-loaded poly(lactic-co-glycolic acid) micro particles that suspended in a gelatine–methacrylate gel inkjet-printed onto titanium coupons [44]. The printing procedure is already validated and the prototype materials are currently being tested. In the case of endotracheal tubes, our current results suggest that all of the tested antimicrobial compounds would be beneficial for the prevention of *S. aureus* adhesion and subsequent biofilm formation, but **DHA1** appears to be the best candidate.

To the best of our knowledge, this is one of the first studies showing a positive effect of novel antimicrobials on the antibiofilm-clearing capacity of immune cells. This is of particular relevance because it does not only provide new alternatives to fight against the immense burden of bacterial biofilms, but it also sets the basis for a new in vitro system to accelerate the drug discovery process, thereby enabling better selection of antimicrobial incorporation into medical devices.

## 5. Conclusions

We showed the suitability of a coculture of *S. aureus* and differentiated HL-60 cells as an in vitro assay to assess the applicability of antimicrobial compounds for the protection of medical devices. As a proof-of-concept, we tested three antimicrobials, concluding that the DHA derivative **DHA1** is the best candidate for incorporation into implantable devices, as it does not only prevent biofilm formation on titanium but also seems to enhance the antibacterial capability of immune cells. On the other hand, according to our results, all of the antimicrobial compounds studied here, i.e., the two DHA derivatives, **DHA1** and **DHA2,** and the flavan derivative, **FLA1**, can tentatively be regarded as promising candidates to form part of anti-infective endotracheal tubes.

## Figures and Tables

**Figure 1 microorganisms-08-01757-f001:**
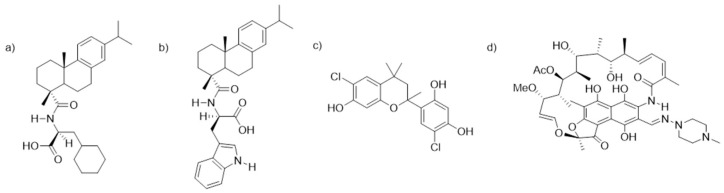
Chemical structures of the two dehydroabietic acid (DHA) derivatives, (**a**) *N*-(abiet-8,11,13-trien-18-oyl)cyclohexyl-L-alanine, (**b**) *N*-(abiet-8,11,13-trien-18-oyl)-D-tryptophan, coded **DHA1** and **DHA2**, the flavan derivative, (**c**) 6-chloro-4-(6-chloro-7-hydroxy-2,4,4-trimethylchroman-2-yl)benzene-1,3-diol, coded **FLA1**, and (**d**) rifampicin, coded **RIF**.

**Figure 2 microorganisms-08-01757-f002:**
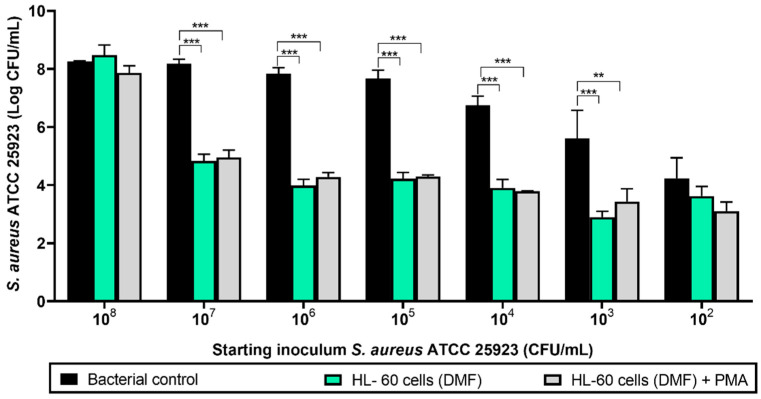
Viable counts of 24-hour-old biofilms formed by different concentrations of *S. aureus* ATCC 25923 cocultured with HL-60 cells on 96-well microplates. Black columns represent the bacterial control (viable attached cells in the absence of HL-60 cells). Green columns show the coculture of *S. aureus* ATCC 25923 with HL-60 cells differentiated with *N*,*N*-dimethylformamide (DMF). Gray columns show the coculture of *S. aureus* ATCC 25923 with HL-60 cells differentiated with DMF and activated with phorbol 12-myristate 13-acetate (PMA). Values are means and SD of three independent experiments (**** p* < 0.001; *** p <* 0.01).

**Figure 3 microorganisms-08-01757-f003:**
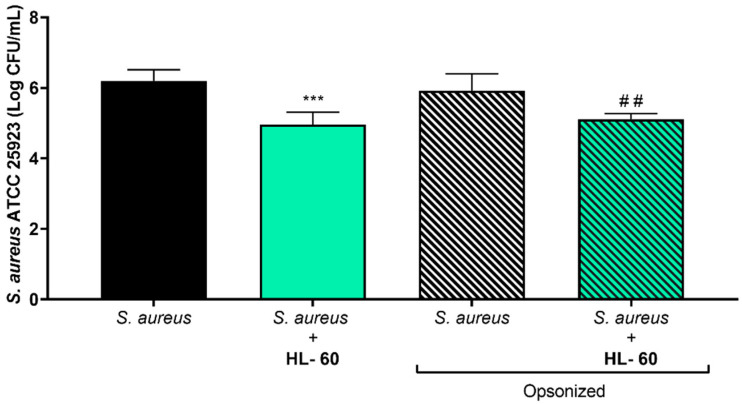
Viable counts of adhered opsonized and nonopsonized *S. aureus* ATCC 25923 on titanium coupons after coculture with HL-60 cells for 24 h. The left half of the graph includes the nonopsonized *S. aureus* ATCC 25923 biofilm formation, in absence (black column) or cocultured with HL-60 cells (green column). The right half includes the opsonized *S. aureus* ATCC 25923 biofilm formation, in absence (white/black column) or cocultured with HL-60 cells (green/black column). “*” indicates statistical differences with the nonopsonized *S. aureus* ATCC 25923 in monoculture, while “#” represents statistical differences with the opsonized *S. aureus* ATCC 25923 in monoculture with Welch’s unpaired *t*-test (**** p* < 0.001; *^##^ p <* 0.01). Values are means and SD of three independent experiments.

**Figure 4 microorganisms-08-01757-f004:**
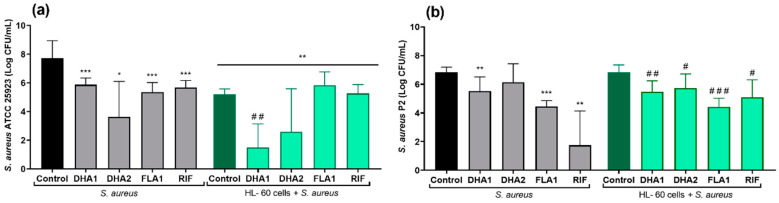
Viable counts of adhered *S. aureus* ATCC 25923 (**a**) and *S. aureus* P2 (**b**) on titanium coupons exposed to different antimicrobial compounds (tested at 50 µM) after 24 h of incubation. The gray bars show the attached viable bacteria when exposed just to the antimicrobial compounds, while the green bars show the results when *S. aureus* strains were cocultured with HL-60 cells. “*” indicates statistical differences with the control in monoculture while “#” represents statistical differences with the control in cocultured controls with Welch’s unpaired *t*-test (** p <* 0.05; *** p* < 0.01; **** p <* 0.001)/ *^#^ p* < 0.05; *^##^ p <* 0.01; *^###^ p<* 0.001). Values are means and SD of three independent experiments.

**Figure 5 microorganisms-08-01757-f005:**
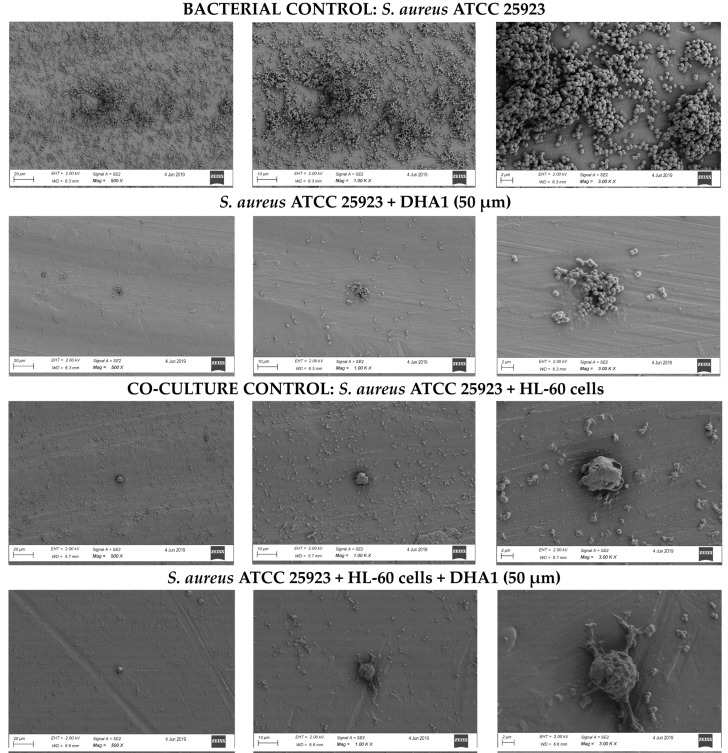
Representative images acquired by SEM. From left to right the same section of the titanium coupon is shown with different magnification, 500, 1000 and 3000×. Upper row of images, bacterial control, i.e., coupons incubated with *S. aureus* ATCC 25923 only. Second row, titanium coupons incubated with *S. aureus* ATCC 25923 and treated with **DHA1**. Third row, cocultured control, titanium coupons incubated with *S. aureus* ATCC 25923 and HL-60 cells simultaneously. Fourth row, titanium coupons cocultured with *S. aureus* ATCC 25923 and HL-60 and treated with **DHA1**.

**Figure 6 microorganisms-08-01757-f006:**
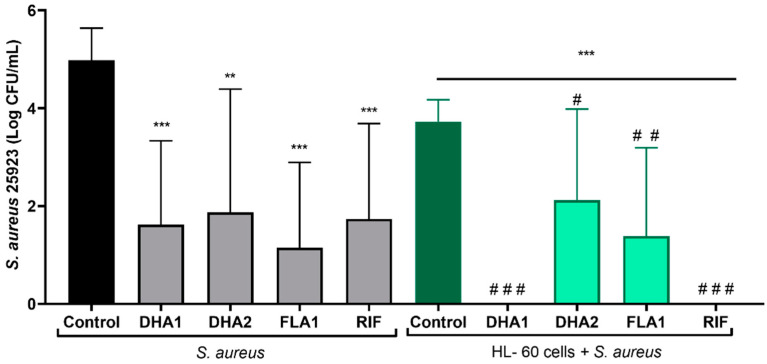
Viable counts of adhered *S. aureus* ATCC 25923 on LDPE tubes exposed to different antimicrobial compounds (tested at 50 µM) after 24 h of incubation. The gray bars show the attached viable bacteria when exposed just to the antimicrobial compounds, while the green bars show the results when *S. aureus* strains were cocultured with HL-60 cells. “*” indicates statistical differences with the control in monoculture, while “#” represents statistical differences with the control in cocultured controls with Welch’s unpaired *t*-test (*** p* < 0.01; *** *p* < 0.001)/(^#^
*p* < 0.05; ^##^
*p* < 0.01; ^###^
*p* < 0.001). Values are means and SD of three independent experiments.

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
