# Peer review of "Combined Effect of Naturally-Derived Biofilm Inhibitors and Differentiated HL-60 Cells in the Prevention of Staphylococcus aureus Biofilm Formation"

_microorganisms, 2020, doi:10.3390/microorganisms8111757_

Round 1
Reviewer 1 Report
This paper is analyzing the anti-biofilm capacity of three naturally-derived biofilm inhibitors when combined with immune cells, in order to assess their applicability in implantable titanium devices and low-density polyethylene (LDPE) endotracheal tubes.
This is interesting, however the anti-biofilm capacity of three naturally-derived biofilm inhibitors should be determined from not only the amounts of biofilm but also the characteristics of biofilm, so that biofilm characterization, such as Raman spectroscopy, is also indispensable.
Author Response
This paper is analyzing the anti-biofilm capacity of three naturally-derived biofilm inhibitors when combined with immune cells, in order to assess their applicability in implantable titanium devices and low-density polyethylene (LDPE) endotracheal tubes.
Point 1: This is interesting, however the anti-biofilm capacity of three naturally-derived biofilm inhibitors should be determined from not only the amounts of biofilm but also the characteristics of biofilm, so that biofilm characterization, such as Raman spectroscopy, is also indispensable.
We agree that Raman spectroscopy would be a very useful method in order to study the effects on biofilms of the compounds included in this manuscript. This technique has been shown to be valuable for analysing the effects of these antimicrobials on the biofilm structure and molecular composition [1-3], but we consider that this information would have been particularly valuable if we would be re-proposing these biofilm inhibitors as treatments of a pre-formed biofilm. In that kind of study, it would be pertinent to use this technique, as it would allow to study the diffusion and distribution[4] processes of these molecules in the biofilm structure as well as analysing what chemical changes on the composition of the extracellular matrix they would cause. Nevertheless, in the current manuscript, we are exploring the utility of these compounds as part of biomaterials so that they can prevent an infection, not eradicate an existing one. We used Scanning Electron Microscopy to confirm (via direct visualization) that in the treated materials the bacterial attachment was significantly reduced, when compared to untreated surfaces. Because of the dramatic reduction on attached bacteria we observed, we could not have really performed an analysis on the composition of the biofilms, especially of the extracellular polymeric substances. We are aware that Raman spectroscopy has also been used to compare the spectra of planktonic cells and their biofilm forms [5], what could also give valuable information in the study of the effects of compounds on biofilm formation and maturation.
However, despite this documented value of Raman spectroscopy for studying in-depth the effect of biofilm inhibitors, the main scope of our manuscript does not reside on performing such characterization, and therefore we don’t think that using Raman spectroscopy is essential in our case. The aim of this paper is to provide with a new methodology to further assess the applicability of antimicrobial compounds as part of medical devices, with the specific antimicrobial compounds we have included serving as proof-of-concepts. An extensive in vitro characterization of the effects of these compounds on biofilm prevention and disruption in microtiter well plates has previously been performed by our research team [6,7]. Based upon those results, we selected the compounds included in this study in order to optimize this new methodology. We will, in any case, take in consideration the reviewer comments, and explore Raman spectroscopy within a future characterization of the effects of these compounds on pre-formed biofilms, on medically-relevant surfaces.
References
1. Ivleva, N.P.; Kubryk, P.; Niessner, R. Raman microspectroscopy, surface-enhanced Raman scattering microspectroscopy, and stable-isotope Raman microspectroscopy for biofilm characterization. Analytical and bioanalytical chemistry 2017, 409, 4353-4375, doi:10.1007/s00216-017-0303-0. 2. Jung, G.B.; Nam, S.W.; Choi, S.; Lee, G.J.; Park, H.K. Evaluation of antibiotic effects on Pseudomonas aeruginosa biofilm using Raman spectroscopy and multivariate analysis. Biomedical optics express 2014, 5, 3238-3251, doi:10.1364/boe.5.003238. 3. Lu, X.; Samuelson, D.R.; Rasco, B.A.; Konkel, M.E. Antimicrobial effect of diallyl sulphide on Campylobacter jejuni biofilms. The Journal of antimicrobial chemotherapy 2012, 67, 1915-1926, doi:10.1093/jac/dks138.
4. Suci, P.A.; Geesey, G.G.; Tyler, B.J. Integration of Raman microscopy, differential interference contrast microscopy, and attenuated total reflection Fourier transform infrared spectroscopy to investigate chlorhexidine spatial and temporal distribution in Candida albicans biofilms. J Microbiol Methods 2001, 46, 193-208, doi:10.1016/s0167-7012(01)00268-8. 5. Kusić, D.; Kampe, B.; Ramoji, A.; Neugebauer, U.; Rösch, P.; Popp, J. Raman spectroscopic differentiation of planktonic bacteria and biofilms. Analytical and bioanalytical chemistry 2015, 407, 6803-6813, doi:10.1007/s00216-015-8851-7. 6. Manner, S.; Skogman, M.; Goeres, D.; Vuorela, P.; Fallarero, A. Systematic Exploration of Natural and Synthetic Flavonoids for the Inhibition of Staphylococcus aureus Biofilms. International journal of molecular sciences 2013, 14, 19434-19451, doi:10.3390/ijms141019434. 7. Manner, S.; Vahermo, M.; Skogman, M.E.; Krogerus, S.; Vuorela, P.M.; Yli-Kauhaluoma, J.; Fallarero, A.; Moreira, V.M. New derivatives of dehydroabietic acid target planktonic and biofilm bacteria in Staphylococcus aureus and effectively disrupt bacterial membrane integrity. European journal of medicinal chemistry 2015, 102, 68-79, doi:10.1016/j.ejmech.2015.07.038.
Reviewer 2 Report
This manuscript describes a trial to test some biofilm inhibitors in their effect on the biofilm prohibition by polymorphonuclear leukocytes (PMN) on two artificial materials: titanium (implantable) and LDPE (endotracheal tube material). To achieve this aim, authors made a new in vitro model combining biofilm assay with PMN cells. The importance of this study’s aim is well explained together with a series of appropriate literatures. The methods are carefully designed and described well with sufficient information. This is a kind of welcomed challenging study, though a primitive first step as authors says, and this reviewer would like to encourage authors to improve the following points prior to the acceptance for publication.
Major points.
- Results and discussion sections are redundant. Similar descriptions are repeated in the result section (of course though it is about different experimental targets), which can be minimized by e.g. combining Figures 4 and 5: which also facilitate readers to see common or the strain specific effects.
- Discussion section needs extensive revision to minimize the redundant and non-relevant description. Please don’t repeat the so far described things in introduction or result without any significant deduction of new insight or hypothesis etc from them. Repeating results and just adding some limitation of the experiments are not attractive discussion. If authors want to describe the limitations of this in vitro system, they can be summarized into one section. Also many future plans gives non-positive (uncompleted) impression to this manuscript, which can also be shortened. These changes would not affect the relevance of the established in vitro system: prerequisite all readers would agree with is that this is in vitro model far from complicated in vivo
- This reviewer assumes the concentration (50µM) of each compounds would be sub-inhibitory levels, but could not find the description of each MIC or MBC values for planktonic and biofilm cells. These information needs to be added somewhere in the manuscript. In the developed assay, how much did the supplementation of each compound affect entire cell growth/survival (planktonic + biofilm)? Please also clarify this point experimentally.
Minor points.
- What would happen if opsonized S. aureus and PMA activated HL-60 were used?
- Lines 462: “it seems DHA2 can synergistically act with the HL-60 cells, as against S. aureus P2 alone it did not reduce the bacterial attachment, but it reduced this attachment when co-cultured with HL-60 cells.”: this comparison is not appropriate according to the data.
- Authors used sonication to disperse the biofilm before counting cfu. Does sonication in the applied condition(10 min at 35kHz) decrease the original cfu value? e.g. if planktonic cells were subjected to this sonication, how much cells can survive? Alternatively, it might be possible to compare percentages of the PI-stained cells in biofilm and after sonication.
- English needs revision. Many typo.
Author Response
This manuscript describes a trial to test some biofilm inhibitors in their effect on the biofilm prohibition by polymorphonuclear leukocytes (PMN) on two artificial materials: titanium (implantable) and LDPE (endotracheal tube material). To achieve this aim, authors made a new in vitro model combining biofilm assay with PMN cells. The importance of this study’s aim is well explained together with a series of appropriate literatures. The methods are carefully designed and described well with sufficient information. This is a kind of welcomed challenging study, though a primitive first step as authors says, and this reviewer would like to encourage authors to improve the following points prior to the acceptance for publication.
Major points.
Point 1: Results and discussion sections are redundant. Similar descriptions are repeated in the result section (of course though it is about different experimental targets), which can be minimized by e.g. combining Figures 4 and 5: which also facilitate readers to see common or the strain specific effects.
We agree with the reviewer, the text has now been modified accordingly, and Figure 4 and 6 merged in one.
Point 2: Discussion section needs extensive revision to minimize the redundant and non-relevant description. Please don’t repeat the so far described things in introduction or result without any significant deduction of new insight or hypothesis etc from them. Repeating results and just adding some limitation of the experiments are not attractive discussion. If authors want to describe the limitations of this in vitro system, they can be summarized into one section. Also many future plans gives non-positive (uncompleted) impression to this manuscript, which can also be shortened. These changes would not affect the relevance of the established in vitro system: prerequisite all readers would agree with is that this is in vitro model far from complicated in vivo.
We agree with the reviewer, the text has now been modified accordingly.
Point 3: This reviewer assumes the concentration (50µM) of each compounds would be sub-inhibitory levels, but could not find the description of each MIC or MBC values for planktonic and biofilm cells. These information needs to be added somewhere in the manuscript. In the developed assay, how much did the supplementation of each compound affect entire cell growth/survival (planktonic + biofilm)? Please also clarify this point experimentally.
We agree with the comment of the reviewer, the text has now been modified accordingly. Data on MIC as well as IC50 values on biofilms of these compounds were already available, as they have been previously characterized by our research team [1,2]. The concentration of 50 µM was chosen as in our previous publication all the three compounds showed, at this concentration, a high efficacy in preventing S. aureus (ATCC 25923 and P2) biofilm formation in 96-well microplates, and none of them displayed any cytotoxicity towards mammalian cells in 96-well microplates [3].
Table 1 previously reported antimicrobial activity and anti-biofilm potencies of the biofilm inhibitors. DHA1 and DHA2 are coded 11 and 9b, respectively in [1] while FLA1 is coded 291 in [2].
Minor points.
Point 1: What would happen if opsonized S. aureus and PMA activated HL-60 were used?
This is an interesting question and we appreciate the reviewer bringing this out. Based upon this question, we have tested the antibacterial effects of PMA activated HL-60 cells against opsonized S. aureus and we did not detect any differences with the other groups (HL-60 cells and PMA activated HL-60 cells against nonopsonized S. aureus)
Point 2.: Lines 462: “it seems DHA2 can synergistically act with the HL-60 cells, as against S. aureus P2 alone it did not reduce the bacterial attachment, but it reduced this attachment when co-cultured with HL-60 cells.”: this comparison is not appropriate according to the data.
This line has now been excluded in the modified manuscript.
Point 3: Authors used sonication to disperse the biofilm before counting cfu. Does sonication in the applied condition (10 min at 35 kHz) decrease the original cfu value? e.g. if planktonic cells were subjected to this sonication, how much cells can survive? Alternatively, it might be possible to compare percentages of the PI-stained cells in biofilm and after sonication.
Planktonic S. aureus was exposed to sonication (10 min at 35 kHz) and compared with the same inoculum without sonication and differences in variability were not observed (Figure 2; now included as SFigure 1 in current publication).
Point 4: English needs revision. Many typo.
As per the reviewer suggestion, we have performed an extensive revision of the manuscript.
References
1. Manner, S.; Vahermo, M.; Skogman, M.E.; Krogerus, S.; Vuorela, P.M.; Yli-Kauhaluoma, J.; Fallarero, A.; Moreira, V.M. New derivatives of dehydroabietic acid target planktonic and biofilm bacteria in Staphylococcus aureus and effectively disrupt bacterial membrane integrity. European journal of medicinal chemistry 2015, 102, 68-79, doi:10.1016/j.ejmech.2015.07.038. 2. Manner, S.; Skogman, M.; Goeres, D.; Vuorela, P.; Fallarero, A. Systematic Exploration of Natural and Synthetic Flavonoids for the Inhibition of Staphylococcus aureus Biofilms. International journal of molecular sciences 2013, 14, 19434-19451, doi:10.3390/ijms141019434. 3. Reigada, I.; Perez-Tanoira, R.; Patel, J.Z.; Savijoki, K.; Yli-Kauhaluoma, J.; Kinnari, T.J.; Fallarero, A. Strategies to Prevent Biofilm Infections on Biomaterials: Effect of Novel Naturally-Derived Biofilm Inhibitors on a Competitive Colonization Model of Titanium by Staphylococcus aureus and SaOS-2 Cells. Microorganisms 2020, 8, doi:10.3390/microorganisms8030345.
Round 2
Reviewer 1 Report
This paper is good enough for publication.
Author Response
We thank the time invested in reviewing our manuscript and the recommendation for its publication in microorganisms
Reviewer 2 Report
Manuscript was corrected extensively and all of the claims from this reviewer was answered in the point-to-point letter. However, it is necessary to edit some minor points prior to the publication.
Minor point
- a) (original major point 3)
I appreciate this table, which helps to think what’s going on in the established experimental system. I hope this tale must be somewhere in this manuscript rather than mentioning previous papers. If length is limited, supplementary figure is also fine.
- b) (original minor point 1)
I wonder if authors could add this figure as supplementary figure. And just shortly mention around line 344.
- c) (related to original minor point 2)
Line 384-387 “Interestingly, DHA1, DHA2, FLA 1, and RIF all significantly reduced the attachment of S. aureus P2 in co-culture (p =0.001, p =0.025, p<0.001, and p=0.029, when comparing, respectively, DHA1, DHA2, FLA1 and RIF with the co-culture control) suggesting an additive effect or synergy between the antimicrobials and the HL-60 cells.”
I think this description is misleading. These compounds has effect in mono-culture, and nothing like additive effect or synergy is seen. This improper understanding seems to be affecting other parts through the manuscript e.g.:
Line 533-535 “ it is only DHA1 treatment that further potentiates the action of HL60 cells …”
It cannot be generalized, because for PS, this is not applicable
Line 645-647 “Compound DHA1 was shown to further potentiate the bacterial clearance caused by HL-60, and such effect was further confirmed by SEM. This further potentiation was also observed for the clinical strain S. aureus P2, but in this case,….”
This is followed by proper interpretation, but this itself must be avoided due to the reason described in the following part.
Line 654 “These results demonstrate that DHA1 seems to further increase the bacterial clearance by HL60,”
same as above
Line 405-406: probably this description here becomes not appropriate when above correction done. This can be deleted here, because same phrases appear multiple times elsewhere.
Author Response
Point 1: a) (original major point 3)
I appreciate this table, which helps to think what’s going on in the established experimental system. I hope this tale must be somewhere in this manuscript rather than mentioning previous papers. If length is limited, supplementary figure is also fine.
We agree with the reviewer, this Table has now been added as supplementary material (STable 1). It has been mentioned in section 2.1.
Point 2: b) (original minor point 1)
I wonder if authors could add this figure as supplementary figure. And just shortly mention around line 344.
This figure has now been added as supplementary material (SFigure 2) and mentioned where the reviewer suggested.
Point 3: c) (related to original minor point 2)
Line 384-387 “Interestingly, DHA1, DHA2, FLA 1, and RIF all significantly reduced the attachment of S. aureus P2 in co-culture (p =0.001, p =0.025, p<0.001, and p=0.029, when comparing, respectively, DHA1, DHA2, FLA1 and RIF with the co-culture control) suggesting an additive effect or synergy between the antimicrobials and the HL-60 cells.”
I think this description is misleading. These compounds has effect in mono-culture, and nothing like additive effect or synergy is seen. This improper understanding seems to be affecting other parts through the manuscript e.g.:
Line 533-535 “ it is only DHA1 treatment that further potentiates the action of HL60 cells …”
It cannot be generalized, because for PS, this is not applicable
Line 645-647 “Compound DHA1 was shown to further potentiate the bacterial clearance caused by HL-60, and such effect was further confirmed by SEM. This further potentiation was also observed for the clinical strain S. aureus P2, but in this case,….”
This is followed by proper interpretation, but this itself must be avoided due to the reason described in the following part.
Line 654 “These results demonstrate that DHA1 seems to further increase the bacterial clearance by HL60,”
same as above
Line 405-406: probably this description here becomes not appropriate when above correction done. This can be deleted here, because same phrases appear multiple times elsewhere.
We agree with the comments of the reviewer and the text has been modified accordingly